# Dual nature of magnetic nanoparticle dispersions enables control over short-range attraction and long-range repulsion interactions

Ahmed Al Harraq[1], Aubry A. Hymel[1], Emily Lin [2], Thomas M. Truskett[2] & Bhuvnesh Bharti [1✉]

Competition between attractive and repulsive interactions drives the formation of complex phases in colloidal suspensions. A major experimental challenge lies in decoupling independent roles of attractive and repulsive forces in governing the equilibrium morphology and long-range spatial distribution of assemblies. Here, we uncover the 'dual nature' of magnetic nanoparticle dispersions, particulate and continuous, enabling control of the short-range attraction and long-range repulsion (SALR) between suspended microparticles. We show that non-magnetic microparticles suspended in an aqueous magnetic nanoparticle dispersion simultaneously experience a short-range depletion attraction due to the particulate nature of the fluid in competition with an in situ tunable long-range magnetic dipolar repulsion attributed to the continuous nature of the fluid. The study presents an experimental platform for achieving in situ control over SALR between colloids leading to the formation of reconfigurable structures of unusual morphologies, which are not obtained using external fields or depletion interactions alone.

[1] Cain Department of Chemical Engineering, Louisiana State University, Baton Rouge, LA 70803, USA. [2] McKetta Department of Chemical Engineering, University of Texas at Austin, Austin, TX 78712, USA. ✉email: bbharti@lsu.edu

Dynamic control over assembly of microscopic building blocks is a key challenge in designing materials which reconfigure in space and time[1,2]. Strategies for engineering the required balance of interparticle attractions and repulsions and in situ tuning of complex interaction potentials remain elusive[3]. Experimental realization of colloidal molecules[4], crystals[5–7] and other mesophases is traditionally focused on the formation of equilibrium structures based on the initial composition and concentration of building blocks[8,9]. Programming assembly pathways[10–12] to form different transient states from the same initial suspension is a standing challenge. External electric, magnetic, and optical fields are among the most prominent tools for directing assembly of colloidal particles with in situ control[13–16]. However, their application is often limited to the introduction of an attractive potential to reversibly transition colloids between assembled and disassembled states[17–20]. A realm of intricate mesophases arises when multiple attractive and repulsive interactions compete, which is widely observed in biological matter[21–23] and is largely untapped in synthetic materials. Developing approaches to direct the structure and dynamics of colloids via competing interactions is key in engineering materials[24,25] with life-like features[26] such as reconfigurability, self-healing[27] and regeneration.

Dispersions of magnetic nanoparticles (MNPs) showcase a variety of reversible assembly mechanisms when exposed to external magnetic fields[28]. In such processes, *discrete particles* acquire magnetic dipoles that align them with the external field according to their axis of magnetization. Polarized particles attract and repel each other in the direction parallel and perpendicular to the external field, respectively[29]. These anisotropic interactions have been exploited to direct the assembly of structures with complex two-dimensional (2D)[30,31] and three-dimensional (3D) morphologies[32–35]. Magnetic fields can also manipulate non-magnetic objects that are dispersed in a magnetic fluid through so-called 'negative magnetophoresis'[36]. In this context, a homogeneous dispersion of MNPs may be approximated as *continuous fluids* with a fixed magnetic susceptibility[37]. Thus, larger non-magnetic colloids suspended in a MNP dispersion behave diamagnetically when exposed to an external magnetic field, i.e., they become polarized with a moment aligned antiparallel to the external field[38]. Such continuum treatment has been theoretically justified and experimentally demonstrated for systems comprising suspended colloids that are two orders of magnitude larger than the MNPs[38,39].

Magnetic interactions in superparamagnetic particles normally only take place when the external field is present and cease once it is removed. This is the basis for programming colloidal materials with two configurations: the assembled state when the external field is present and the disassembled state when it is absent. Embedding additional configurations within the same initial dispersion is far from trivial because it requires control over a dynamic energy landscape beyond the on/off states. To address this, we take advantage of the 'dual nature', discrete and continuous, of MNP dispersions in which larger microparticles are suspended. On one hand, the discrete nature of nanoparticles induces an osmotic pressure pushing larger suspended colloids together. This depletion attraction is known to take place when the distance between microparticle surfaces is smaller than the diameter of nanoparticles[40,41]. Sufficiently strong short-range attraction promotes the assembly of microparticles into 2D crystals. On the other hand, the continuous nature of the magnetic fluid allows to 'polarize' suspended microparticles which would otherwise be non-magnetic[38]. The interaction between magnetized particles is effectively a long-range repulsion between side-to-side dipoles in the plane orthogonal to the external magnetic field.

The simultaneous dual functionality of MNP dispersions grants a control over the assembly of suspended colloids. Using a fixed binary suspension of $Fe_3O_4$ nanoparticles and polystyrene (PS) microparticles, we investigate the role of tuning the balance of attraction and repulsion in structuring colloidal matter in a quasi-2D plane. When attraction dominates, we observe the formation of large hexagonal colloidal crystals. Conversely, strong repulsion impedes assembly and maintains particles in a disordered state. When the magnitude of depletion attraction and magnetic repulsion are balanced, we observe the assembly of discrete clusters, where the size and morphology is governed by the strength of the applied magnetic field. The pathways of assembly and disassembly are summarized in Fig. 1.

Cluster formation results from a delicate balance of short-range attraction and long-range repulsion referred to as SALR[22,42–46]. The assemblies continually break, merge, and reconfigure. while dissipating external magnetic energy input via the electromagnet, in a unique class of *dynamic*-SALR (*d*-SALR). The cluster state can be considered a pseudo-equilibrium phase where the input of energy via an electromagnet is a pre-requisite. Clusters of equal size and different configurations occur in a form of colloidal isomerism that reveals a preference for higher-symmetry morphologies, in contrast with previous findings[47]. The versatility of the MNP dispersion lies in the ability to not only program the strength of the depletion attraction, but also to modify the magnetic repulsion in situ. This allows to fine tune the interaction energy landscape while observing the response of microparticles. A unique feature of MNP dispersions is the potential to construct *d*-SALR models using colloids that are not magnetic.

## Results

**Experimental details**. All experiments are done with the same binary suspension of microparticles and magnetic nanoparticles, placed in an electromagnetic Helmholtz coil that generates a uniform magnetic field tunable in the range 0–10 kA m$^{-1}$. The individual building blocks for assembly are fluorescent PS

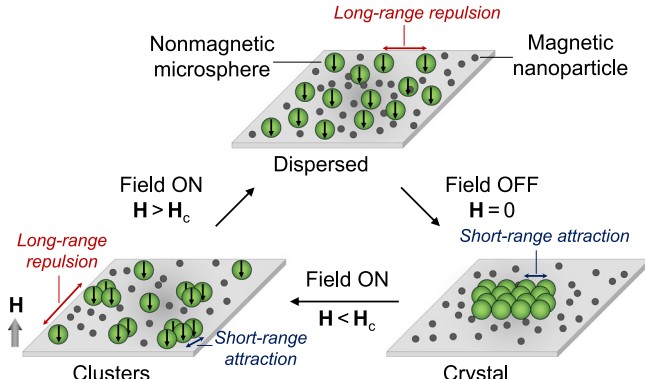

**Fig. 1 Schematic of microparticle assembly in magnetic nanoparticle dispersion.** A suspension of microparticles immersed in a MNP dispersion undergoes a cycle of assembly and disassembly. When no external magnetic field is applied ($H = 0$), the discrete nature of the nanoparticle dispersion induces a short-range depletion attraction between the microsphere which assemble into crystal structures. When the external magnetic field is applied with a magnitude lower than a certain critical value ($H < H_c$), the microparticles become polarized due to the continuous nature of the MNP dispersion: a long-range magnetic repulsion ensues in competition with the short-range depletion attraction causing the assembly of small microparticle clusters. The black arrow within a microparticle represents the dipole moment vector. At relatively high field strength ($H > H_c$), the magnetic repulsion between microparticles overcomes the depletion attraction and acts to maintain a disordered state.

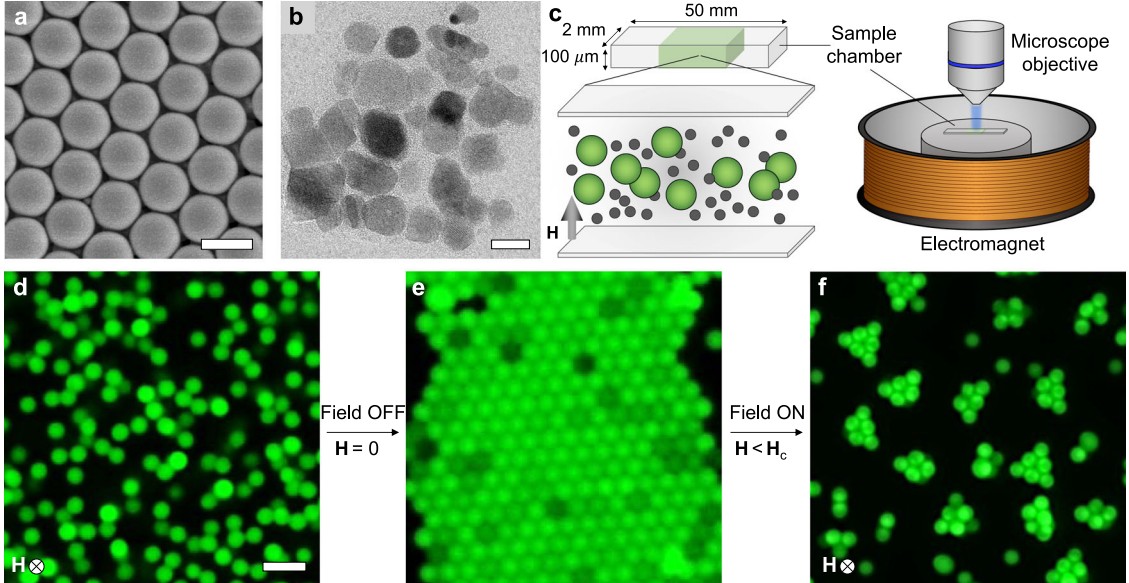

**Fig. 2 Experimental setup and main observations. a** Scanning electron microscopy image of the PS microparticles used. Scale bar: 2 μm. **b** Transmission electron microscopy image of $Fe_3O_4$ nanoparticles responsible for the short-range depletion attraction and long-range magnetic repulsion responsible for assembly of the PS microparticles. Scale bar: 20 nm. **c** Schematic of the experimental setup comprising a sample chamber placed in the center of an electromagnetic coil under the microscope objective. The sample is the binary suspension of PS microparticles and $Fe_3O_4$ nanoparticles enclosed in a flat glass capillary which is sealed at both ends. **d**–**f** Fluorescence microscopy images of unassembled microparticles, assembled crystal and clusters. Scale bar: 5 μm.

microparticles (Magsphere Inc.) with a 1 μm radius (Fig. 2a and Supplementary Fig. 1). Their surface is prefunctionalized with carboxylate groups which induce a net negative charge on the particles (Supplementary Fig. 2). The MNPs are an aqueous dispersion of $Fe_3O_4$ nanoparticles (Ferrotec) with an average radius of 8 nm (Fig. 2b and Supplementary Fig. 3). The MNP dispersion contains a proprietary anionic surfactant to prevent nanoparticle aggregation. The PS microparticles (0.1 vol%) and $Fe_3O_4$ NPs (2 vol%) are suspended in water at pH 6 containing 40 mM NaCl which is added to screen electrostatic repulsion. The ζ-potential of the PS microparticles and the $Fe_3O_4$ NPs is, respectively, −25 and −50 mV, under experimental conditions. The suspension is sealed inside a flat glass capillary (Vitrocom) using ultraviolet (UV) curable glue. This sample chamber is then transferred to the center of an electromagnetic Helmholtz coil setup which is connected to an external power supply. As direct current is passed through the coil, the associated magnetic field is generated with strength **H** proportional to the current. The Helmholtz coil containing the sample chamber is placed under an upright microscope for live imaging during assembly both with and without exposure to the magnetic field. A schematic of the experimental setup is given in Fig. 2c.

**Assembly and disassembly of colloidal crystals.** In the absence of a magnetic field, MNPs in the dispersion trigger the excluded volume interactions among the larger colloids. Thus, the microparticles initially in a disordered state (Fig. 2d) spontaneously rearrange to form a hexagonal colloidal crystal (Fig. 2e). At the PS and MNP concentrations used, the microparticles tend to form a single layer of this ordered state. Applying an external magnetic field activates the repelling function of the MNP dispersion. Its effects depend on the magnitude of the field strength $H_c$: if $H > H_c$, the crystal is fully disassembled back into its individual components. Conversely, if $H < H_c$, short-range attraction and long-range repulsion compete in the assembly of dynamic clusters (Fig. 2f), where $H_c$ is the critical field strength. Note that the

observed phase transition upon increasing **H** is gradual, and the purpose behind using $H_c$ is merely to provide a qualitative dependence of assemblies on applied magnetic field strength.

We characterize the assembly and disassembly of colloidal crystals with the hexagonal bond order parameter, $\psi_6$, which quantifies the degree of hexatic (six-fold orientational) order of a structure in a range from 0 to 1[48]. Briefly, $\psi_{6,j}$ indicates the local crystallinity of a single particle $j$ with respect to the orientation of its nearest neighbors:

$$\psi_{6,j} = \frac{1}{N_{C,j}} \sum_{k=1}^{N_{C,j}} e^{i6\theta_{jk}} \tag{1}$$

where $N_{C,j}$ is the number of microparticles within a diameter range from the center of $j$ and $\theta_{jk}$ is the angle between the bond segment connecting $j$ with each neighboring particle $k$. The global hexagonal bond order parameter, $\langle\psi_6\rangle = \frac{1}{N_t}\sum\psi_{6,j}$, indicates the degree of crystallinity in the assembly by averaging all individual $\psi_{6,j}$ over the total number of particles $N_t$. A perfect 2D hexagonal lattice has $\langle\psi_6\rangle = 1$ while total disorder is indicated by $\langle\psi_6\rangle \sim N_b^{-1/2}$, where $N_b$ is the total number of nearest-neighbor bonds in the system. The weak short-range attraction between microparticles leads to the assembly and growth of hexagonal crystals near the bottom surface of the sample chamber. Introducing a relatively strong repulsion with a magnetic field causes the melting-like disassembly of crystals starting from the edges and any defects present (Fig. 3a, b and Supplementary Movie 1). This indicates that as **H** increases, the long-range repulsion between microparticles in the 2D plane of assembly increases and overcomes the short-range depletion attraction. We tracked the response of microparticles to the external fields with video microscopy and found that $\psi_6$ decreases with time upon the application of the magnetic field. We find that the $\psi_6$ decreases in a nearly linear way with the strength of the external magnetic field (Fig. 3c). Starting from an assembled state with $\langle\psi_6\rangle \approx 0.9$, hexatic order is reduced to 0.75 and 0.43 when

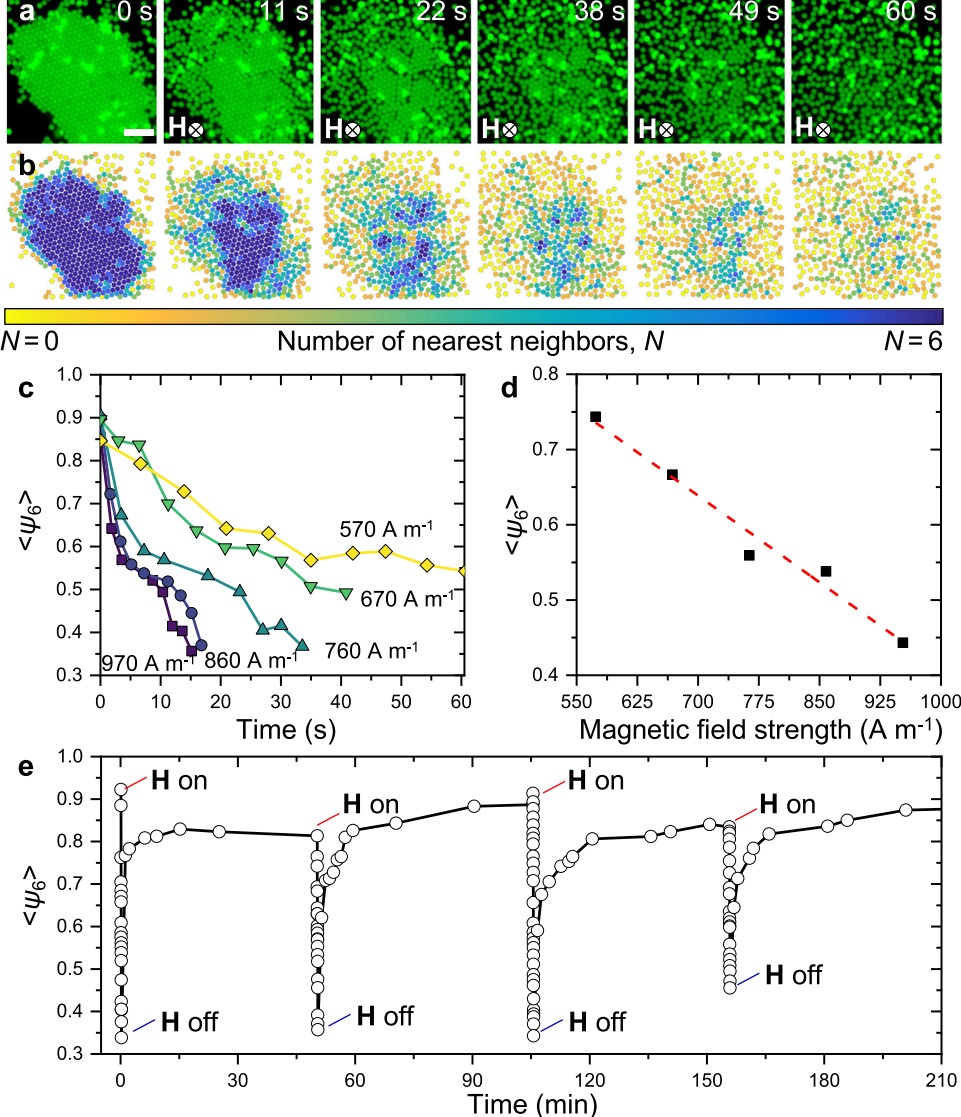

**Fig. 3 Assembly and disassembly of colloidal crystals. a** Time sequence of frames showing the disassembly of a colloidal crystal at **H** = 550 A m$^{-1}$ and **b** corresponding color maps of the number of nearest neighbors, $N$. The disassembly occurs within 60 s starting from the edge and defects where crystal bonds are weaker. Scale bar: 10 μm. **c** Decrease in global crystallinity quantified by $\langle\psi_6\rangle$ for different magnetic field strengths. The values of the **H** (in A m$^{-1}$) are provided next to the corresponding curve. **d** The rate of disassembly decreases linearly with increasing magnetic field strength. All values of $\langle\psi_6\rangle$ are computed after 10 s from initiating disassembly. **e** Cycles of order–disorder transitions correspond to the assembly of crystals via depletion attraction and their disassembly by introducing an external magnetic field.

the external field is applied for 10 s with **H**, respectively, 570 and 970 A m$^{-1}$ (Fig. 3d). When removing the magnetic field, crystal assembly resumes and $\langle\psi_6\rangle$ increases back to >0.8 after which disassembly may be reinitiated in a cycle of order–disorder transitions that is virtually infinite and controlled solely by the external magnetic field (Fig. 3e, Supplementary Fig. 4 and Supplementary Movie 2).

**Theoretical background**. All observed phenomena emerge from a balance of electrostatic, van der Waals, depletion and magnetic interactions taking place in the quasi-2D geometry. Microparticles in aqueous solution experience Derjaguin–Landau–Verwey–Overbeek (DLVO) interactions which combine electrostatic repulsion and van der Waals attraction (see Supplementary Note 1). The net outcome of electrostatic forces is a repulsion between particles due to the similar charging of counterion double-

layers (Supplementary Fig. 5)[49]. By adding NaCl ions, we reduce the Debye length to ~1.5 nm to minimize the role of DLVO interactions in all experiments. As a result, the net effect of DLVO forces is minimal compared with the interactions arising from the properties of the MNPs. The driving force for assembly is the depletion attraction induced by the discrete nature of MNP dispersions. Conversely, the driving force for disassembly is the magnetic repulsion induced by the continuous nature of the suspension medium. A schematic of the key interactions at play is provided in Fig. 4a. Attraction occurs as depletant NPs crowd the volume adjacent the microparticles, the so-called depletion zone with a thickness equal to the diameter of the NPs. Overlap of depletion zones causes an imbalance in the distribution of NPs around the microparticles, which results in a net osmotic force pushing colloids together. The depletion interaction energy $U_d$ scales with the number density of depletants $\rho$ and the size of the

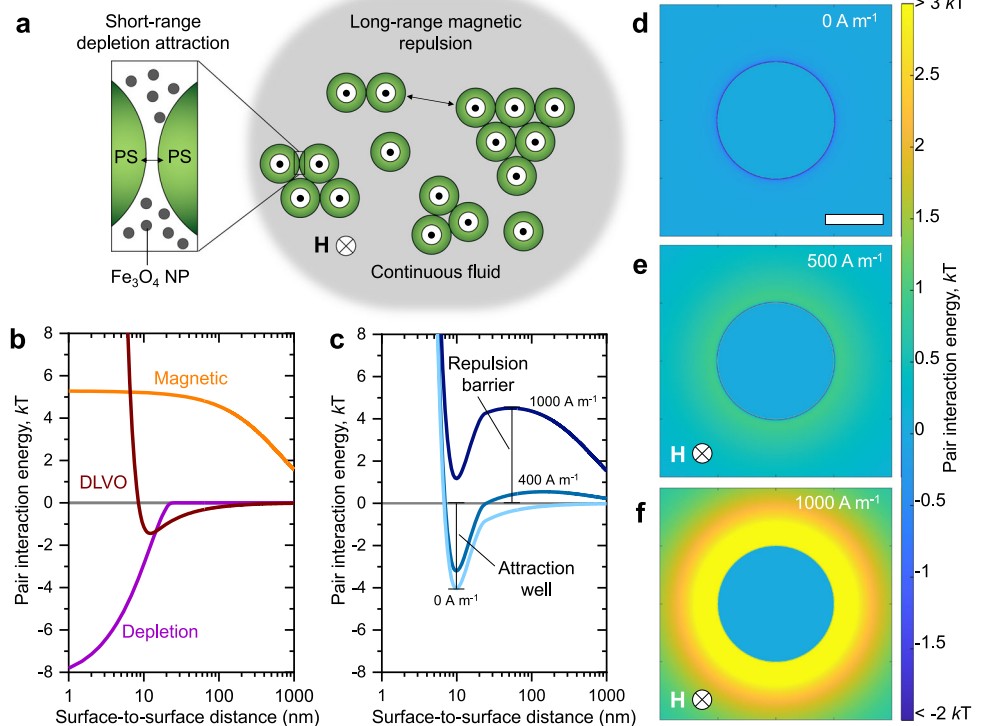

**Fig. 4 Theoretical background. a** Schematic of the discrete and continuous nature of MNP dispersions. Individual $Fe_3O_4$ NPs act as depletants while the magnetic fluid renders microparticles diamagnetic in an out-of-plane external magnetic field. **b** Pair interaction energy contributions of electrostatic-van der Waals (DLVO), depletion and magnetic interactions. **c** Plots of total pair interaction energy with differing contribution of magnetic repulsion. As the field strength increases, the net interaction switches from attractive to repulsive indicating the role of the magnetic contribution in morphing the energy landscape in favor of assembly or disassembly. **d**–**f** Color maps representing the distribution of interaction energy surrounding a microparticle with an out-of-plane field intensity respectively 0, 500, and 1000 A m$^{-1}$. Scale bar: 1 μm.

excluded volume, as expressed by the Asakura-Oosawa potential[40]:

$$U_d(D) = -\frac{\pi}{6}\rho kT(2r - D)^2\left(3R + 2r + \frac{D}{2}\right) \text{ for } 0 \le D \le 2r \quad (2)$$

$$U_d(D) = 0 \text{ for } D > 2r \quad (3)$$

where $D$ is the surface-to-surface distance between microparticles, $R$ is their radius, and $r$ is the radius of the MNPs. In Eq. (2), $k$ is Boltzmann's constant and $T$ is the temperature. Due to the adsorption of surfactant on its surface, the effective depletant size is increased by an estimated surfactant bilayer thickness which has been previously shown to be ~4 nm[50]. The driving force for disassembly is the magnetic repulsion between particles positioned in the same plane (i.e., the bottom surface of the sample chamber) and polarized in the orthogonal plane by the external field.

By approximating microparticles as point-dipoles, we compute the magnetic interaction energy $U_m$ between particles $i$ and $j$ as the following[51]:

$$U_m(s) = \frac{\mu_0/4\pi}{s^3}\left(\mathbf{m}_i \cdot \mathbf{m}_j - 3\frac{(\mathbf{m}_i \cdot \mathbf{s})(\mathbf{m}_j \cdot \mathbf{s})}{s^2}\right) \quad (4)$$

where $s = |\mathbf{s}|$ is the center-to-center distance between two polarized microparticles $i$ and $j$. The magnetic dipole moment is given for each particle as $\mathbf{m} = 4\pi R^3 K_{CM}\mathbf{H}$, thus scaling with its volume, the field intensity and with the real part of the Clausius-Mossotti function $K_{CM}$. The $K_{CM}$ term incorporates the fundamental role of the MNP dispersion as a continuous medium with bulk magnetic susceptibility $\chi_m \approx 0.4$ (Supplementary Fig. 6) in which non-magnetic particles with susceptibility $\chi_i = 0$ are immersed:

$$K_{CM} = \frac{\chi_i - \chi_f}{\chi_i + 2\chi_f + 3} \quad (5)$$

The numerator of Eq. (5) is the difference in magnetic susceptibility between particle and medium. Since the microparticles in use are non-magnetic, the gradient, and thus $\mathbf{m}$, acquire a negative value. This is associated with a diamagnetic response to external magnetic fields such that all microparticles in the system acquire dipoles antiparallel to the direction of the applied external field.

We estimated the contributions of DLVO, depletion and magnetic interactions to the total pair potential using MATLAB. Addition of 40 mM NaCl to the dispersion results in screening of the double layer repulsion barrier. This is critical to enable depletion attraction, which is operational only at distances shorter than the diameter of the MNP. The resulting shallow attraction well (~1 $kT$) in the DLVO interactions did not lead to the aggregation of microparticles (Fig. 4b, and Supplementary Fig. 5). Summing together all the contributions, we reveal the energy landscape of two interacting particles. By modulating the strength of the external magnetic field, the net pair potential helps in clarifying the observed assembly phenomena (Fig. 4c). On one hand, at $\mathbf{H} = 0$ A m$^{-1}$, the energy landscape around each particle presents a minimum representing an attraction well of ~4 units of thermal energy $kT$ induced by the depletion potential. On the other hand, at $\mathbf{H} = 1000$ A m$^{-1}$, the magnetic field introduces a ~4 $kT$ repulsion barrier at ~100 nm from the particle surface which causes disassembly and prevents assembly of microparticles in plane. Interestingly, there is a range of external field strength that leads to the simultaneous formation of an attraction

well at 10 nm and a broad repulsion barrier at ~100 nm, e.g., at $H = 400$ A m$^{-1}$. Given the anisotropy of magnetic interactions, it is necessary to note that the behavior described is limited to microparticles interacting in a 2D plane orthogonal to the direction of the magnetic field. Interactions in this plane are reasonably summarized by the total pair potential described, which may be expanded to color maps showing the regions of attraction and repulsion surrounding a microparticle with an out-of-plane magnetic field (Fig. 4d–f and Supplementary Fig. 7). In addition, we note that in the range of field intensities used, there is minimal tendency of both the $Fe_3O_4$ nanoparticles and the PS microparticles to assemble into chains within the duration of applied field (Supplementary Figs. 8 and 9). Chaining is only observed for microparticles at higher magnetic fields which we do not reach in any experiment described (Supplementary Fig. 10).

**Cluster formation in *dynamic*-SALR.** The competition between short-range attraction and long-range repulsion between non-magnetic microparticles is finely tuned by experimentally controlling the strength of magnetic field, giving rise to a dynamic form of SALR. We find that the 0–500 A m$^{-1}$ range of field strengths does not fully overcome the depletion attraction allowing to observe the effects of competition. In this complex energy landscape, microparticles are simultaneously pushed together by depletion interaction and pulled apart by the magnetic field: the result is the formation of dynamic clusters that occasionally interact, merge, split and exchange particles. Similar clustering was previously observed in systems of active particles showcasing mobility-induced phase separation[52–54]. In this case, no active motion is present and the clustering results from the competition between depletion attraction and magnetic repulsion.

After several hours of exposure to weak fields, we observe a self-limiting behavior of the assembled clusters which continue exchanging particles yet do not appear to change in size

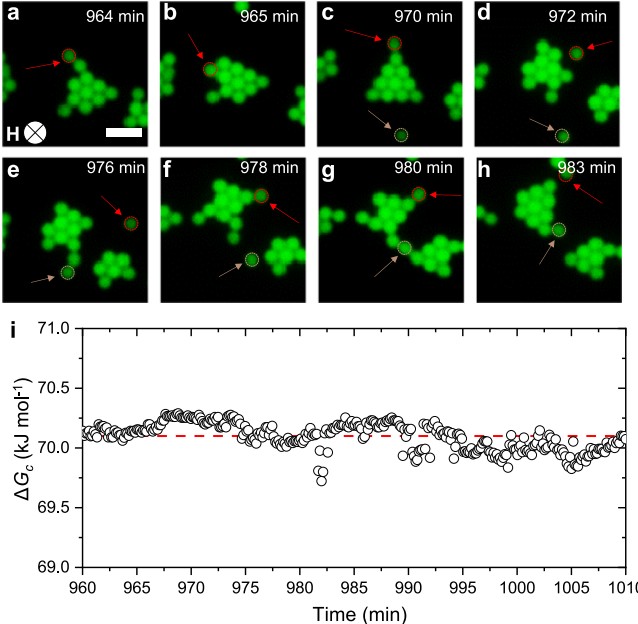

**Fig. 5 Dynamic cluster formation. a–h** Time-sequence showcasing typical rearrangement of individual microparticles in dynamic clusters over 20 min, with $H = 290$ A m$^{-1}$. Scale bar: 3 μm. **i** Change in Gibbs free energy of clusters over 50 minutes, calculated from image analysis obtained after 12 h of equilibration at $H = 290$ A m$^{-1}$. The dashed line is a linear fit showing a mean value of 70.1 kJ mol$^{-1}$.

(Fig. 5a–h and Supplementary Movie 3). Treating these competition-driven assemblies as association colloids in analogy with surfactant self-assembly, we estimate their molar Gibbs free energy change, $\Delta G_c = -R_g T \ln C_c + \bar{n} R_g T \ln C_p$[49]. Here, $R_g$ is the universal gas constant, $T$ is temperature, $C_c$ and $C_p$ are, respectively, the concentration of assembled clusters and mono-mer particles measured from recorded videos. We find that $\Delta G_c$ oscillates around a constant value in the tens of kilojoules per mole (Fig. 5i), which is comparable with values found for other association colloids such as micelles[55]. The result indicates that the assembled clusters, while continuously rearranging and exchanging particles, are in fact in a pseudo-equilibrium. This is a steady state that is only rendered possible by dissipating external magnetic energy. Without the contribution of the magnetic dipolar repulsion to the interaction energy landscape, the same particles assemble into the ordered crystal state shown in Fig. 2e.

Cluster formation was theorized to be a generic feature of at least two competing interactions, one of which promotes aggregation while the other powers interparticle separation[56].

Accordingly, the typical size of clusters is expected to decrease as the intensity of the repulsive component increases. The d-SALR here shown is driven by an energy-dissipative repulsion that prevents the equilibrium crystallization which is entropically favored by the depletion attraction.

As the strength of the magnetic field is increased, the size of the clusters decreases as shown in Fig. 6a–c. Each $H$ is associated with a specific energy landscape and a distribution of cluster sizes, $n$, which broadens at lower field strengths where crystallites of different shapes and sizes are allowed to form (Fig. 6d, Supplementary Fig. 11 and Supplementary Table 1).

We observe the formation of colloidal isomers in clusters of size 6 and higher. There are different configurations of equal-sized 2D structures such as the triangle, parallelogram and chevron arrangements observed for $n = 6$. We count the occurrence of all isomers of $n = 6$ in suspensions exposed to magnetic fields of strength between 300 and 400 A m$^{-1}$, after allowing equilibration for 12 h (Fig. 7a). We find that the triangle isomer occurs with 77% probability while the fraction for the parallelogram and chevron shapes are, respectively, 12% and 11% (Fig. 7b and Supplementary Figs. 12 and 13). Thus, the highest symmetry configuration is exceedingly more common which is in stark contrast with previous studies that found hexamer colloidal clusters to exist in a triangle state for one third of the time spent in the parallelogram and chevron states[47,57]. The total repulsion among dipoles within each distinct configuration is approximately equal (Supplementary Fig. 14). Given the identical excluded volumes and surface energies across three isomers, one would expect to encounter parallelogram and chevron states more often than the triangular one. However, our experiments show an opposite population distribution with the largest fraction occupied by the triangular isomer. While the origin of our experimental observation is unknown, we speculate that the long-range repulsion[38,58] between clusters could drive preferential formation of symmetric triangular structures over asymmetric chevrons and parallelograms. The uniform separation between the clusters can be clearly observed in the experimental image shown in Fig. 7a, which indicates the presence of long-range intercluster repulsions. Further investigation is necessary to identify the nature and role of cluster–cluster repulsion on the equilibrium distribution of cluster morphologies with dissimilar symmetries. Regardless of the origin, the predominance of the triangular configuration demonstrates the ability of d-SALR to enable and preserve assembled states of colloidal particles which are otherwise inaccessible by field-induced or depletion interactions alone.

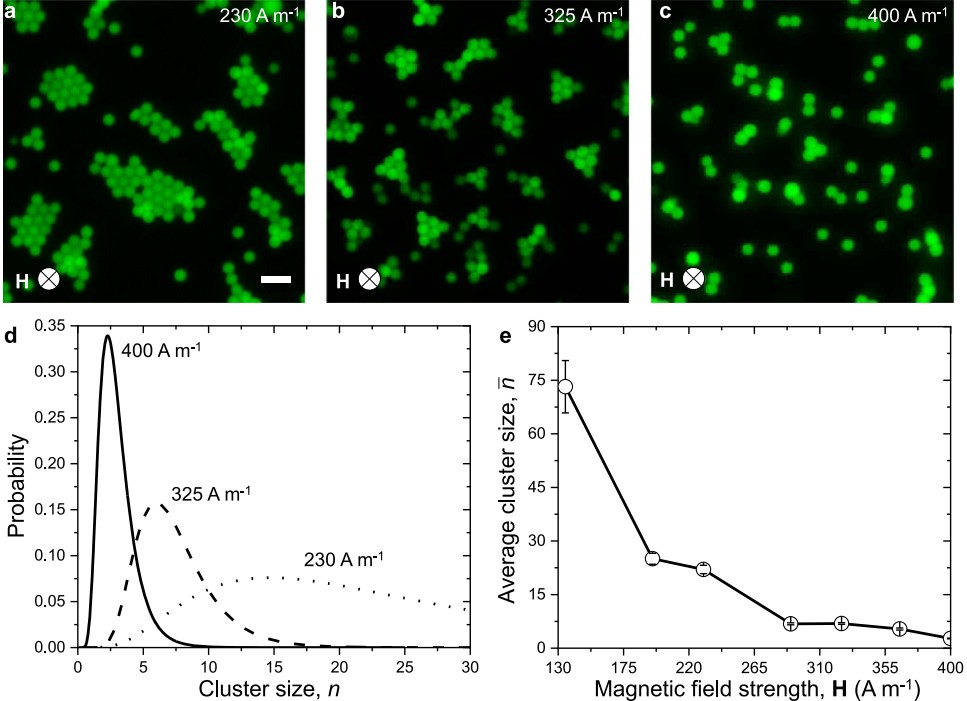

**Fig. 6 Control and distribution of cluster size.** Colloidal clusters assembled via depletion attraction and magnetic repulsion with magnetic field strengths of **a** 230 A m$^{-1}$, **b** 325 A m$^{-1}$, and **c** 400 A m$^{-1}$. Scale bar: 5 μm. **d** Distribution of cluster sizes at different magnetic field strengths. The y-axis indicates the probability of microparticles to be assembled in a cluster of size n. The value of the applied magnetic field, **H** (A m$^{-1}$) is provided next to the corresponding curve. **e** Average cluster size decreases with increasing strength of the external magnetic field. Error bars indicate the experimental standard deviation.

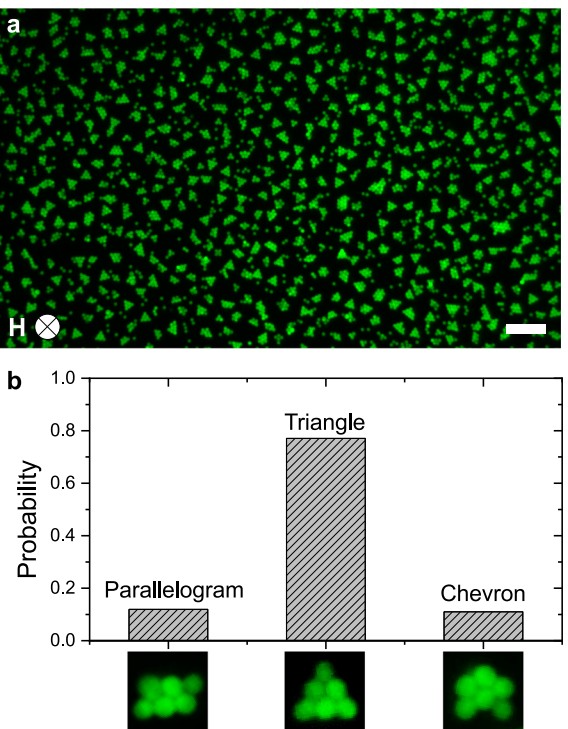

**Fig. 7 Formation of isomeric clusters. a** Dynamic clusters assembled with **H** = 365 A m$^{-1}$. Scale bar: 20 μm. **b** Measured frequency of occurrence of isomers for $n = 6$, with **H** = 325 A m$^{-1}$. Fluorescence images show the measured shapes, i.e., from left to right, the parallelogram, triangle, and chevron states.

Magnetic nanoparticle dispersions simultaneously drive attraction and repulsion between larger colloids providing dynamic control over assembled structures using an external magnetic field. On one hand, the discrete particulate nature of the nanoparticles induces short-range depletion attraction which promotes formation of 2D hexagonal colloidal crystals. On the other hand, the continuous nature of magnetic dispersion offers a way to tune long-range repulsion which favors cluster disassembly. The attraction–repulsion balance can be controlled dynamically to guide the assembly of pseudo-equilibrium clusters with programmable average size and morphology. This experimental model opens the door for design of clusters and other mesophases starting from simple components and using inverse methods[59,60]. The ability to engineer interaction energy landscapes[61] and morph them in situ may be useful in understanding the role of competing and cooperating interactions in dynamic phenomena, such as crystal nucleation[62], gelation[63] and the glass transition[64].

## Methods

**Materials and sample preparation.** Carboxylated polystyrene microparticles were purchased from Magsphere Inc., 2.0 μm in diameter and labeled with green fluorescent dye (catalog number: CAF-002UM). The MNPs used are a dispersion of $Fe_3O_4$ nanoparticles containing a proprietary anionic surfactant (Ferrotec EMG 705). Flat borosilicate capillaries used to contain the suspension were purchased from Vitrocom (VitroTubes$^{TM}$ 5010). Prior to use, capillaries were treated with a commercially available polysiloxane solution (Rain-X) to render their surfaces hydrophobic. The treatment involved immersing the capillaries in the solution for one hour and subsequently flushing the contents out with air. After being filled with the colloidal suspension, each end of the capillaries was sealed using UV sensitive glue (Loctite AA 349) which was cured for 60 s in a UV flood curing system (Uvitron International Intelliray).

**Magnetic field experiments.** A magnetic field was externally generated using a custom-built electromagnet made from a commercially available air-core solenoid

(TEMCo Industrial 14 AWG Copper Magnet Wire; 36 m long, 0.08 cm thick, ~450 turns). The coil was fit on a microscope stage and connected to an external direct current generator (BK Precision 1665) forming a vertical Helmholtz coil setup that provides a uniform magnetic field to the sample placed in its center. The electromagnet was operated using the external power supply in the current controlled mode which allows to tune the exact strength of the associated magnetic field as measured by a gaussmeter (AlphaLab Inc. GM2).

**Microscopy.** All fluorescence imaging and video recording was done using a Leica DM6 upright microscope equipped with a DFC9000 GTC camera and a EL 6000 fluorescence light source. The objective used was a Leica ×40/0.55 air objective and the filter cube was green fluorescent protein. Scanning electron microscopy was done using a Quanta 3D DualBeam FEG FIB-SEM with an accelerating voltage of 5 kV. The sample imaged was a droplet of the microsphere stock suspension, which was dried on carbon tape and coated with a 5 nm layer of platinum to prevent charging. Transmission electron microscopy of the MNPs was done using a JEOL JEM2010 operated at 200 kV.

**Image analysis.** Analysis of each individual image or video frame was done using the ImageJ software package[65]. The Trackmate plugin[66] was used to extrapolate particle coordinates, which were subsequently used to calculate $\psi_{6,j}$ and $\langle\psi_6\rangle$ and plot color maps in MATLAB. Analysis of cluster size and concentration was done by binarizing frames into particulate objects whose area is measured using the 'Analyze particles' function.

**SQUID magnetometry.** Measurement of the magnetic properties of the MNP dispersion was done using a MPMS XL SQUID magnetometer (Quantum Design). The sample was inserted in a 5 mm × 5 mm × 5 mm volume within a plastic enclosing. Changes in the magnetic flux are created and detected by sliding the sample through a superconducting coil.

## Data availability
Supplementary Movie 1, 2 and 3 are available for download. The datasets generated and/or analyzed during the current study are available from the corresponding author on reasonable request.

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

## Acknowledgements

B.B. acknowledges the financial support from the National Science Foundation (NSF) under Grants CBET−1943986 (NSF-CAREER) and CBET−2038305. T.M.T. acknowledges support from the Welch Foundation (Grant No. F-1696).

## Author contributions

B.B. and A.A. conceived the project and planned the experiments. A.A. and A.A.H. performed the experiments under the guidance of B.B. E.L. and T.M.T. assisted with analysis and interpretation of the isomerism in cluster states. A.A. performed experimental data analysis. B.B. and A.A. wrote the manuscript. All authors read and commented on the manuscript.

## Competing interests

The authors declare no competing interests.
