## [Peer review file · Communications Chemistry]

Reviewers' comments:

Reviewer #1 (Remarks to the Author):

Review of 'Competitive interactions via the 'dual nature' of magnetic nanoparticle dispersions' by Harraq et al..

This is a very interesting paper that demonstrates in-situ tuneable SALR competition between microparticles, or colloids. This is achieved, for the first time as far as I know, using a magnetic fluid with immersed non-magnetic microparticles. The magnetic fluid is a suspension of dispersed magnetic nanoparticles with a narrow size distribution which induce both short-range depletion attraction and long-range negative magnetophoretic repulsion in the much larger suspended microparticles. In this 2-D case, this leads to the self-assembly of discrete 2-D clusters.

The manuscript is mostly fine, and I recommend publication after some alterations to the description of the SALR cluster fluid, where some inaccuracy is apparent. There are also a few other cases where greater accuracy with the language is warranted.

In fact, the behaviour of an SALR fluid is sensitive to both the competition between attractive and repulsive interactions, AND to the magnitude of the attractive and repulsive interactions. For relatively weak interactions we expect SALR fluids to be very dynamic with rapid exchange of particles between clusters and the dispersed phase. This is normal, and seen in, for example, the Monte Carlo simulations by Sweatman, Fartaria and Lue (JCP vol. 140, p 124508 (2014)) as well as in much theoretical work. Such SALR fluids can be said to be at equilibrium. Whether such equilibrium SALR fluids can be realised experimentally is another issue.

However, as the short-range attraction becomes ever shorter its magnitude must increase in order to cause aggregation. In such cases the dynamics slow down due to the size of the attractive barrier, relative to the thermal energy, that must be overcome to break a bond between neighbouring particles. As dynamics slow down, we can expect eventually to enter a non-equilibrium regime where the particles become frustrated relative to the timescale of the experiment. Such SALR fluids might be described as 'non-dynamic', although the more usual language is 'frustrated'.

Moreover, if the long-range repulsion is also quite strong, relative to the thermal energy, as well as quite long-ranged compared to the particle size, then a disordered Wigner glass can result. In this case, dynamics are slowed due to the magnitude of repulsive interactions, not attractions. Either way, dynamics can be slowed such that non-equilibrium, or frustrated, states are achieved. This is often the case in experiments, but not so much in simulation or theoretical work where interactions can be tailored arbitrarily.

In essence, I think it is usual to assume SALR fluids are 'dynamic', or at equilibrium, and it is important to say if they are not, in which case they are usually described as 'frustrated', or non-equilibrium.

In line with this clarification, the authors should consider modifying the following sections of their manuscript;

1. Page 3 'Such short-range attraction promotes the assembly of microparticles into 2D crystals.'

This is true only for sufficiently strong and short-ranged attractions. It is not true for weaker and longer-ranged short-range attractions where liquid-like clusters are also possible.

2. Page 3 'The interaction between magnetized particles is effectively a long-range repulsion between side-to-side dipoles in the plane orthogonal to the external magnetic field.'

This a key phrase and the concept is not immediately obvious to the reader. I recommend the authors insert a diagram showing the orientation of the magnetic dipoles relative to the applied field and the 2-D setup.

3. Page 4, 'we observe the assembly of discrete clusters, where the size and morphology is governed

by the strength of the applied magnetic field.'

While this is true, being the only means of in-situ control, there are other ex-situ experimental parameters than can be controlled, and it would be very interesting to understand their effects. Indeed, the concentration of the MNPs is crucial, as it controls both the strength of the depletion attraction and the magnetophoretic repulsion. And, as the authors point out later, cluster morphology might also be dependent on microparticle concentration. In fact, this is known to be true (see Sweatman, Fartaria and Lue again, who present a cluster fluid phase diagram). Indeed, their work shows that the transition between a dispersed SALR state and a SALR cluster fluid is always continuous, even if occurs over a narrow range of phase space. So, the authors of this manuscript should be careful with their language concerning 'transitions', 'thresholds', 'critical parameters', and so on.

4. Page 4, 'The assemblies continually break, merge, and reconfigure while dissipating external magnetic energy, in a unique class of dynamic-SALR (d-SALR).'

It is not obvious this is correct. In any case, such 'dynamic' fluids are usually described as 'equilibrium', as explained earlier. In fact, at equilibrium there will not be any dissipation, as motion is thermal, i.e. Brownian for the MNPs and microparticles. Dissipation can occur on the approach to equilibrium or if there are hydrodynamic forces present, but the authors have not shown this.

5. Page 9, 'The net outcome of DLVO forces is a repulsion between particles due to the similar charging of counterion double-layers.'

And yet, in Figure 4 the DLVO interaction is shown as attractive at long range. Which is correct?

6. Page 9, equations (2) and (3), the parameter 'r' is not described. Is it the radius of MNPs?

7. Page 10, 'We calculated the contributions of DLVO...'

Actually, they are estimated, not calculated. It is possible to measure them by performing an experiment without MNPs at low microparticle concentrations. Then, the resulting microparticle rdf can be converted directly to a DLVO interaction.

8. Page 12, 'giving rise to a dynamic form of SALR.'

Again, 'equilibrium', not 'dynamic', is preferred. Unless, that is, the system has not yet reached equilibrium, or experiences hydrodynamic interactions. But how can we know from the results presented?

9. Page 13, 'The result indicates that the assembled clusters, while continuously rearranging and exchanging particles, are in fact in a pseudo-equilibrium. This is a steady state that is only rendered possible by dissipating external magnetic energy.'

Again, this is not obviously true. Thermal motion alone can cause continuous exchange of particles between clusters and the dispersed phase for an equilibrium SALR fluid. Dissipation is only expected for non-equilibrium fluids. The continuous exchange of particles does not tell us which is true in this case.

10. Page 14, 'one of which promotes crystallization'

Actually, it promotes aggregation, which might lead to crystallisation in cases where the short-range attraction is very short and strong.

11. Page 14, 'The d-SALR shown here is driven by an energy-dissipative repulsion'

Same as above, this has not been shown and might not be true.

12. Page 15, 'we speculate that the long-range repulsion between clusters could drive preferential formation of symmetric triangular structures over asymmetric chevrons and parallelograms.'

This is an interesting point. Essentially, the authors are saying the system is crowded, which is also suggested by Figure 7a. In fact, this system might be a Wigner glass. Can the authors show whether it is a Wigner glass?

Reviewer #2 (Remarks to the Author):

The work of Al Harraq et al describes a magnetic nanoparticle and micron-sized colloid particles mixture. The presence of nanoparticles in the suspension makes the suspension paramagnetic therefore results in diamagnetic behavior of non-magnetic colloids in the presence of a magnetic field. The same nanoparticles exert depletion forces to the larger colloids causing an attraction between these colloids. The authors mention that they use this dual nature of the magnetic nanoparticles to induce short range attractions due to depletion interactions and simultaneously to induce long-range repulsions over an external magnetic field application. The authors claim that the use of competing attractive and repulsive forces to obtain and tune well defined colloid assemblies are not common but the following work (doi.org/10.1103/PhysRevX.5.021012) for instance is a good example of such attractive and repulsive competing interactions and result in well-defined clusters. The work of Erb et al which is Ref 37 in the manuscript can also be considered in this category. Authors should consider this and similar work and consider changing their claims. The assembly shown is in 2D and also the long-range repulsion is also in 2D due to the angle dependence of the dipolar interactions. Therefore, the claims at the beginning are slightly stronger than what authors demonstrate. Still the experiments and the content is robust and the demonstrations are well supported by the experiments and analyses. The paper's novelty is limited in demonstration of depletion interactions together with dipolar interactions. In this context, these two interactions compete and provide a method to control assembly of colloidal clusters. The work and analyses are interesting but not for a broad audience. There have been numerous reports in the last decade or two, which showed many ways of creating and controlling colloidal clustering. Few to name are doi.org/10.1038/nature11564 and DOI: [10.1126/science.1086189](https://doi.org/10.1126/science.1086189).

The work has to be revised considering the comments above and the detailed ones below.

1- The 2D crystals are forming only on the substrate or also in bulk? The depletion forces seem to be weaker than the dipolar forces therefore the assembly seems to be limited at the substrate. Please clarify this point.

2- Is the depletion triggered assembly happening in 3D? A 3D image or a xz scan showing the cross section would help the reader to assure the behavior. Else describe in words/ or with supplementary images.

3-When H is much larger than H_c do the authors observe chain formation? They don't mention/consider this scenario. Please describe such scenario's presence or absence. If not describe why and how?

Reviewer #3 (Remarks to the Author):

The manuscript presents some interesting experiments, supplemented by some calculations, of non-magnetic PS colloids dispersed in an aqueous ferrofluid, in order to study their behavior subject to the application and removal of a uniform magnetic field. The particles, charged, have been dispersed in a solution containing a sufficiently high concentration of electrolytes to screen the electrostatic repulsions, so as not to interfere with the other interactions, namely depletion interactions and magnetic interactions, which are the topic of the work. The authors show that, by fine tuning the concentration of ferrofluids, the PS particles arrange into colloidal crystals induced by attractive depletion interactions due to the ferrofluid nanoparticles. However, the application of an external magnetic field induces repulsive dipolar interactions among the particles, when the field is perpendicular to the plane of observation. By tuning the intensity of the applied field, intermedia situations could be observed, such as the formation of small clusters with fairly regular (i.e., crystalline) structure.

The work is well written, with clear experiments, which have been complemented by calculations showing the relative importance of the various interactions acting on the PS particles. The work is also original, and of broad interest to the community of people working with colloids/soft matter. The work is in my opinion worth publishing. I have a couple of comments that the authors should address.

1) The authors should discuss in some detail whether it still makes sense to use Asakura-Oosawa theory to describe the effect of depletion interactions due to magnetic nanoparticles when a magnetic field is applied. In fact, in this case some formation of chains among the magnetic nanoparticles should be present, which might have a strong effect on the strength of depletion interactions. It is in fact well known that rod-like particles exert stronger attractive depletion interactions than spherical particles because of the larger excluded volume.

2) It would be interesting to see what happens to the particles in the direction of the applied magnetic field, where, on top of the attractive depletion interactions, dipolar interactions are also attractive. Did the authors observe the formation of chain-like clusters in the presence of an applied field?

Response to Reviewers' comments on 'Competitive interactions via the 'dual nature' of magnetic nanoparticle dispersions'

Reviewer 1

General comment 1: This is a very interesting paper that demonstrates in-situ tuneable SALR competition between microparticles, or colloids. This is achieved, for the first time as far as I know, using a magnetic fluid with immersed non-magnetic microparticles. The magnetic fluid is a suspension of dispersed magnetic nanoparticles with a narrow size distribution which induce both short-range depletion attraction and long-range negative magnetophoretic repulsion in the much larger suspended microparticles. In this 2-D case, this leads to the self-assembly of discrete 2-D clusters. The manuscript is mostly fine, and I recommend publication after some alterations to the description of the SALR cluster fluid, where some inaccuracy is apparent. There are also a few other cases where greater accuracy with the language is warranted.

Response: We thank the reviewer for the positive assessment of our work and the careful inspection of the manuscript. Following, is a point-by-point response to the comments offered with details on the actions taken to improve the article.

General comment 2: In fact, the behaviour of an SALR fluid is sensitive to both the competition between attractive and repulsive interactions, AND to the magnitude of the attractive and repulsive interactions. For relatively weak interactions we expect SALR fluids to be very dynamic with rapid exchange of particles between clusters and the dispersed phase. This is normal, and seen in, for example, the Monte Carlo simulations by Sweatman, Fartaria and Lue (JCP vol. 140, p 124508 (2014)) as well as in much theoretical work. Such SALR fluids can be said to be at equilibrium. Whether such equilibrium SALR fluids can be realised experimentally is another issue.

Response: We agree with the Reviewer's definition of terminology regarding the dynamics of particle exchange and the equilibrium conditions underlying SALR fluids. In fact, we have some initial observations of the slowing down of dynamics upon increasing the magnitude of the short-range attraction (not reported in the manuscript). Our use of the terms *pseudo-equilibrium* and *dynamic* regarding the cluster assemblies comes from the role of the magnetic field. Traditionally, SALR clusters have been observed in experiments that balance depletion attraction with electrostatic double layer repulsion: in these cases, the interactions may lead to equilibrium. We effectively replaced the role of electrostatic repulsion with a magnetic dipolar repulsion, i.e., an interaction originating from an external dissipative energy source. The clusters observed in our experiments require continuous input of external magnetic energy, thus we termed the apparent formation of a cluster phase a *pseudo-equilibrium*. In the revised manuscript we clarify this on page 4 by introducing following sentence:

"The cluster state can be considered a pseudo-equilibrium phase where the input of energy via an external magnetic field is a pre-requisite."

General comment 3: However, as the short-range attraction becomes ever shorter its magnitude must increase in order to cause aggregation. In such cases the dynamics slow down due to the size of the attractive barrier, relative to the thermal energy, that must be overcome to break a bond between neighbouring particles. As dynamics slow down, we can expect eventually to

enter a non-equilibrium regime where the particles become frustrated relative to the timescale of the experiment. Such SALR fluids might be described as ‘non-dynamic’, although the more usual language is ‘frustrated’.

Moreover, if the long-range repulsion is also quite strong, relative to the thermal energy, as well as quite long-ranged compared to the particle size, then a disordered Wigner glass can result. In this case, dynamics are slowed due to the magnitude of repulsive interactions, not attractions. Either way, dynamics can be slowed such that non-equilibrium, or frustrated, states are achieved. This is often the case in experiments, but not so much in simulation or theoretical work where interactions can be tailored arbitrarily. In essence, I think it is usual to assume SALR fluids are ‘dynamic’, or at equilibrium, and it is important to say if they are not, in which case they are usually described as ‘frustrated’, or non-equilibrium. In line with this clarification, the authors should consider modifying the following sections of their manuscript;

Response: We agree with the Reviewer’s prediction of the roles of attraction and repulsion. In fact, we observe slowing of the dynamics in more *extreme* conditions of short-range attraction or long-range repulsion. Our initial observations point to the possibility of formation of aggregates in highly attractive potential energy landscapes, and vice versa, formation of Wigner glasses in highly repulsive conditions (not reported in the manuscript). We are dedicating this current article to results observed through a more delicate balance of attraction and repulsion. Characterization of the frustrated cases require a different set of experiments and measurements of dynamics which we intend to carry out and publish in the future.

Comment 1: Page 3 ‘Such short-range attraction promotes the assembly of microparticles into 2D crystals.’ This is true only for sufficiently strong and short-ranged attractions. It is not true for weaker and longer-ranged short-range attractions where liquid-like clusters are also possible.

Response: We agree with the Reviewer’s clarification regarding the requirements for assembly. We amended the sentence on page 3 as follows:

‘Sufficiently strong short-range attraction promotes the assembly of microparticles into 2D crystals.’

Comment 2: Page 3 ‘The interaction between magnetized particles is effectively a long-range repulsion between side-to-side dipoles in the plane orthogonal to the external magnetic field.’ This a key phrase and the concept is not immediately obvious to the reader. I recommend the authors insert a diagram showing the orientation of the magnetic dipoles relative to the applied field and the 2-D setup.

Response: The geometry of the magnetic field is indeed crucial. To provide more clarity on the orientation of the magnetic dipoles in our experiments, we modified the schematic in Figure 1 as follows:

Comment 3: Page 4, ‘we observe the assembly of discrete clusters, where the size and morphology is governed by the strength of the applied magnetic field.’ While this is true, being the only means of in-situ control, there are other ex-situ experimental parameters than can be controlled, and it would be very interesting to understand their effects. Indeed, the concentration of the MNPs is crucial, as it controls both the strength of the depletion attraction and the magnetophoretic repulsion. And, as the authors point out later, cluster morphology might also be dependent on microparticle concentration. In fact, this is known to be true (see Sweatman, Fartaria and Lue again, who present a cluster fluid phase diagram). Indeed, their work shows that the transition between a dispersed SALR state and a SALR cluster fluid is always continuous, even if occurs over a narrow range of phase space. So, the authors of this manuscript should be careful with their language concerning ‘transitions’, ‘thresholds’, ‘critical parameters’, and so on.

Response: We acknowledge that changing the concentration of microparticles and/or depletant affects the assembly. We purposely focus our attention to in-situ tuning of particle interactions because of the unique ability to observe the various assembly states using a single colloidal suspension, at fixed concentration of microparticles, nanoparticles, surfactant, and salt.

The short-range attraction achieved with the selected concentrations (0.1 vol% PS microparticles, 2 vol% Fe_3O_4 NPs) is high enough to induce assembly of the large particles into quasi-ordered large crystal aggregates. Upon applying the external magnetic field, we introduce an additional repulsive interaction that breaks such aggregates into a disordered state. Thus, we refer to an order-disorder transition triggered by relatively strong dipolar repulsion.

In reference to the formation of clusters of different size, we agree with the Reviewer that the process is continuous. Although we identified an approximate range of valid field strengths for cluster formation, we modified our language regarding critical field as follows:

Page 7: 'Note that the observed phase transition upon increasing H is gradual, and the purpose behind using H_c is merely to provide a qualitative dependence of assemblies on applied magnetic field strength.'

Page 6: '*Its effects depend on the magnitude of the field strength*'

Page 13: '*We find that the 0-500 A m⁻¹ range of field strengths does not fully overcome the depletion attraction allowing to observe the effects of competition*'

Comment 4: Page 4, 'The assemblies continually break, merge, and reconfigure while dissipating external magnetic energy, in a unique class of dynamic-SALR (d-SALR).' It is not obvious this is correct. In any case, such 'dynamic' fluids are usually described as 'equilibrium', as explained earlier. In fact, at equilibrium there will not be any dissipation, as motion is thermal, i.e. Brownian for the MNPs and microparticles. Dissipation can occur on the approach to equilibrium or if there are hydrodynamic forces present, but the authors have not shown this.

Response: We agree with the Reviewer that all dynamics at equilibrium are non-dissipative and only thermally driven. Our use of the term dissipative refers to the reliance of our form of SALR to the continuous energy input of the external magnetic field. In the absence of this energy-dissipative term, our experiments display equilibrium in which the microparticles aggregate into many large clusters, i.e., the ordered state shown in Fig. 2e. We have clarified this aspect on page 4 of the manuscript (also see response to General comment 2).

Comment 5: Page 9, 'The net outcome of DLVO forces is a repulsion between particles due to the similar charging of counterion double-layers.' And yet, in Figure 4 the DLVO interaction is shown as attractive at long range. Which is correct?

Response: We thank the Reviewer for the comment. In the original manuscript the statement was referred to the stability of microparticles in the absence of any additional electrolyte. We have now clarified this aspect and point to the new supplementary figure 5 in the statement.

In our assembly experiments we suppress the electrostatic repulsion between the microparticles in order to facilitate the depletion attraction with a low concentration of magnetic nanoparticles. We increased and fixed the concentration of added electrolyte (NaCl) to 40 mM, which resulted in highly screened short-range interactions between microparticles. The resulting depth of attraction well in the DLVO interactions as shown in Fig. 4b is $\sim 1kT$ which did not lead to the aggregation of microparticles. We verified this aspect experimentally (see SI fig. 5), and further clarified in the manuscript by introducing following text:

Page 10: '*The net outcome of electrostatic forces is a repulsion between particles due to the similar charging of counterion double-layers (Supplementary Fig. 5)*'

Page 11: 'Addition of 40 mM NaCl to the dispersion results in screening of the double layer repulsion barrier. This is critical to enable depletion attraction, which is operational only at distances shorter than the diameter of the MNP. The resulting shallow attraction well ($\sim 1kT$) in the DLVO interactions did not lead to the aggregation of microparticles (Fig. 4b, and Supplementary Figure 5).'

Supplementary Figure 5. (A) Micrograph of particles suspended for more than 1 hour in 40 mM NaCl solution, without any nanoparticle depletant. Scale bar: 20 μm . (B) Plots of DLVO interaction between PS microparticles without the addition of salt (red line) and in 40 mM NaCl solution (black line). Adding electrolyte screens electrostatic double layer repulsion to subsequently enable the short-range excluded volume effects of the depletant.

Comment 6: Page 9, equations (2) and (3), the parameter ‘ r ’ is not described. Is it the radius of MNPs?

Response: We thank the Reviewer for identifying the missing definition of the parameter which is indeed the radius of the depletant MNPs.

We added the definition in the revised manuscript on page 10:

$$U_d(D) = -\frac{\pi}{6}\rho kT (2r - D)^2 \left(3R + 2r + \frac{D}{2}\right) \text{ for } 0 \leq D \leq 2r \quad (2)$$

$$U_d(D) = 0 \text{ for } D > 2r \quad (3)$$

where D is the surface-to-surface distance between microparticles, R is their radius, and r is the radius of the MNPs.’

Comment 7: Page 10, ‘We calculated the contributions of DLVO...’ Actually, they are estimated, not calculated. It is possible to measure them by performing an experiment without MNPs at low microparticle concentrations. Then, the resulting microparticle rdf can be converted directly to a DLVO interaction.

Response: We agree with the reviewer that net interaction energy can be obtained by rdf of the microsphere. However, such estimation using experimental images can be challenging with several undesired artifacts. Moreover, such detailed estimation is beyond the scope of our

current manuscript focused on the magnetically controlled SALR. We corrected the indicated statement on page 11 as follows:

'We estimated the contributions of DLVO, depletion and magnetic interactions to the total pair potential using MATLAB (Fig. 4b)'

Comment 8: Page 12, 'giving rise to a dynamic form of SALR.' Again, 'equilibrium', not 'dynamic', is preferred. Unless, that is, the system has not yet reached equilibrium, or experiences hydrodynamic interactions. But how can we know from the results presented?

Response: The observed form of SALR can only be seen while the suspension is exposed to the external magnetic field. We thus infer that the cluster state is a pseudo-equilibrium from the observation that, without the magnetic dipolar repulsion, the particles assemble into ordered clusters. We added the following sentence to the manuscript to highlight and clarify this point, in page 15:

'Without the contribution of the magnetic dipolar repulsion to the interaction energy landscape, the same particles assemble into the ordered state shown in Fig. 2e.'

Comment 9: Page 13, 'The result indicates that the assembled clusters, while continuously rearranging and exchanging particles, are in fact in a pseudo-equilibrium. This is a steady state that is only rendered possible by dissipating external magnetic energy.' Again, this is not obviously true. Thermal motion alone can cause continuous exchange of particles between clusters and the dispersed phase for an equilibrium SALR fluid. Dissipation is only expected for non-equilibrium fluids. The continuous exchange of particles does not tell us which is true in this case.

Response: We agree that thermal motion could, under the right conditions, cause rearrangement of particles between clusters. This is not the case in our system because at the selected concentration of microparticles and depletant, the short-range attraction is too strong to allow rearrangement. Instead, without the external magnetic field, the microparticles eventually assemble into large static crystals. Once the dipolar repulsion is triggered by the external magnetic field, the clusters form with a new energy landscape that allows continuous exchange of particles.

Comment 10: Page 14, 'one of which promotes crystallization' Actually, it promotes aggregation, which might lead to crystallisation in cases where the short-range attraction is very short and strong.

Response: The Reviewer is correct in pointing to the exact role of the attraction. We corrected the sentence (on page 16 now) as follows:

'Cluster formation was theorized to be a generic feature of at least two competing interactions, one of which promotes aggregation while..'

Comment 11: Page 14, 'The d-SALR shown here is driven by an energy-dissipative repulsion' Same as above, this has not been shown and might not be true.

Response: As mentioned above, we believe that this is shown by using the exact same suspension of micro- and nanoparticles. The same particles undergo aggregation without external magnetic energy input. Only when exposed to the dissipative energy source, the clusters form and display rearrangements as discussed.

Comment 12: Page 15, 'we speculate that the long-range repulsion between clusters could drive preferential formation of symmetric triangular structures over asymmetric chevrons and parallelograms.' This is an interesting point. Essentially, the authors are saying the system is crowded, which is also suggested by Figure 7a. In fact, this system might be a Wigner glass. Can the authors show whether it is a Wigner glass?

Response: Indeed, there is a possibility that under certain conditions of external magnetic field, a Wigner glass of clusters may form. We do observe the formation of Wigner glass in our initial experiments (not reported in the manuscript). We intend to look at this aspect of the work by measuring the mean square displacement of particles from our microscopy data. We are working on those experiments currently, which are still proving challenging, but we plan to publish such aspects of the work in a subsequent publication.

Reviewer 2

General comment: The work of Al Harraq et al describes a magnetic nanoparticle and micron-sized colloid particles mixture. The presence of nanoparticles in the suspension makes the suspension paramagnetic therefore results in diamagnetic behavior of non-magnetic colloids in the presence of a magnetic field. The same nanoparticles exert depletion forces to the larger colloids causing an attraction between these colloids. The authors mention that they use this dual nature of the magnetic nanoparticles to induce short range attractions due to depletion interactions and simultaneously to induce long-range repulsions over an external magnetic field application. The authors claim that the use of competing attractive and repulsive forces to obtain and tune well defined colloid assemblies are not common but the following work (doi.org/10.1103/PhysRevX.5.021012) for instance is a good example of such attractive and repulsive competing interactions and result in well-defined clusters. The work of Erb et al which is Ref 37 in the manuscript can also be considered in this category. Authors should consider this and similar work and consider changing their claims. The assembly shown is in 2D and also the long-range repulsion is also in 2D due to the angle dependance of the dipolar interactions. Therefore, the claims at the beginning are slightly stronger than what authors demonstrate. Still the experiments and the content is robust and the demonstrations are well supported by the experiments and analyses. The paper's novelty is limited in demonstration of depletion interactions together with dipolar interactions. In this context, these two interactions compete and provide a method to control assembly of colloidal clusters. The work and analyses are interesting but not for a broad audience. There have been numerous reports in the last decade or two, which showed many ways of creating and controlling colloidal clustering. Few to name are doi.org/10.1038/nature11564 and DOI:10.1126/science.1086189. The work has to be revised considering the comments above and the detailed ones below.

Response: We thank the Reviewer for reading and providing useful comments on our manuscript. We acknowledge that there is existing literature on the design of interactions for colloidal clustering. However, an experimental model with in-situ tunable SALR was never reported to the best of our knowledge. This differs from purely competing long-ranged interactions used for self-limiting assemblies, or with other assembly schemes that rely on multicomponent building blocks or directional bonding. Our interest in such SALR interactions stems from their occurrence in biological matter, especially in protein solutions, which are of both fundamental and technological relevance. At the same time, SALR is a relatively simple phenomenological model with only symmetric pair-interactions yet complex consequences in assembly. To address this point and clarify the difference with the colloidal clustering literature, we modified the following in page 17:

'This experimental model opens the door for design of clusters and other mesophases starting from simple components and using inverse methods'

Comment 1: The 2D crystals are forming only on the substrate or also in bulk? The depletion forces seem to be weaker than the dipolar forces therefore the assembly seems to be limited at the substrate. Please clarify this point.

Response: Our sample configuration promotes assembly in 2D purely from depletion interactions. The dipolar forces are introduced out-of-plane, i.e., perpendicular to the plane of

crystal formation. Thus, the magnetic interaction is repulsive and acts to disassemble the crystals. We changed the schematic of Figure 1 to help clarify this point.

Comment 2: Is the depletion triggered assembly happening in 3D? A 3D image or a xz scan showing the cross section would help the reader to assure the behavior. Else describe in words/ or with supplementary images.

Response: We perform experiments in conditions that limit assembly as much as possible to a single 2D layer. Increasing microparticle concentrations leads to the formation of multilayered colloidal crystals which behave differently and require a different analytical treatment. We added the following sentence to page 7 to clarify this point

'At the PS and MNP concentrations used, the microparticles tend to form a single layer of the ordered state.'

Comment 3: When H is much larger than H_c do the authors observe chain formation? They do not mention/consider this scenario. Please describe such scenario's presence or absence. If not describe why and how?

Response: In the range of fields used, we do not observe any chaining which we only start seeing at relatively high fields $H \sim 2000 \text{ A m}^{-1}$. We added the following sentence on page 12 and Supplementary Figure 10:

'Chaining is only observed for microparticles at higher magnetic fields which we do not reach in any experiment described (Supplementary Fig. 10).'

'Supplementary Figure 10. (A) Fluorescence micrograph and (B) schematic of out-of-plane chaining of microparticles observed when applying a magnetic field with $H = 2000 \text{ A m}^{-1}$. Scale bar: $10 \mu\text{m}$.'

Reviewer 3

General comment: The manuscript presents some interesting experiments, supplemented by some calculations, of non-magnetic PS colloids dispersed in an aqueous ferrofluid, in order to study their behavior subject to the application and removal of a uniform magnetic field. The particles, charged, have been dispersed in a solution containing a sufficiently high concentration of electrolytes to screen the electrostatic repulsions, so as not to interfere with the other interactions, namely depletion interactions and magnetic interactions, which are the topic of the work. The authors show that, by fine tuning the concentration of ferrofluids, the PS particles arrange into colloidal crystals induced by attractive depletion interactions due to the ferrofluid nanoparticles. However, the application of an external magnetic field induces repulsive dipolar interactions among the particles, when the field is perpendicular to the plane of observation. By tuning the intensity of the applied field, intermedia situations could be observed, such as the formation of small clusters with fairly regular (i.e., crystalline) structure.

The work is well written, with clear experiments, which have been complemented by calculations showing the relative importance of the various interactions acting on the PS particles. The work is also original, and of broad interest to the community of people working with colloids/soft matter. The work is in my opinion worth publishing. I have a couple of comments that the authors should address.

Response: We thank the Reviewer for the positive assessment of our work. We address the specific comments as detailed below.

Comment 1: The authors should discuss in some detail whether it still makes sense to use Asakura-Oosawa theory to describe the effect of depletion interactions due to magnetic nanoparticles when a magnetic field is applied. In fact, in this case some formation of chains among the magnetic nanoparticles should be present, which might have a strong effect on the strength of depletion interactions. It is in fact well known that rod-like particles exert stronger attractive depletion interactions than spherical particles because of the larger excluded volume.

Response: The Reviewer raises an interesting point regarding the validity of the Asakura-Oosawa potential for non-spherical depletants. We estimated the magnetic dipolar interaction between the magnetic nanoparticles used (Supplementary Figure 8), based on our SQUID measurement of the dipole moment of those particles. We find that, in the range of fields used, the moment induced in each particle is too weak to induce assembly. Below is the Supplementary Figure 8 showing that the attraction energy between two magnetic nanoparticles is <0.5 kT even under fields stronger than we used in experiments.

While we do not anticipate any significant chaining of magnetic nanoparticles in our experimental conditions of low field strengths (up 1000 A/m), it would be very interesting to manipulate the strength of the depletion attraction by changing the morphology and chaining state of the depletants in-situ.

Comment 2: It would be interesting to see what happens to the particles in the direction of the applied magnetic field, where, on top of the attractive depletion interactions, dipolar interactions are also attractive. Did the authors observed the formation of chain-like clusters in the present of an applied field?

Response: Indeed, the magnetic interaction used in our experiments can lead to attraction instead of repulsion. We purposely limit the strength of the magnetic field to values at which we do not observe any significant chaining as estimated in the following Supplementary Figure 9:

At higher magnetic fields, we do in fact observe some chaining indicating a systemic change in the energy landscape from competing to cooperating interactions (see Supplementary Figure 10). However, in the current manuscript we specifically focus on low field strengths where such chaining is absent.

Reviewers' comments:

Reviewer #1 (Remarks to the Author):

The authors have addressed the review comments well. However, they should explain why their system is continuously dissipating energy, as this seems to be at the heart of the confusion surrounding terminology. Once the magnetic field is applied, where is the continuing energy source? A permanent magnet generates a magnetic field without expending energy, I think. Perhaps their magnetic field is generated by an electromagnet, which does continuously expend energy, but this is a different issue.

Reviewer #2 (Remarks to the Author):

Response to general comment: We thank the Reviewer for reading and providing useful comments on our manuscript. We acknowledge that there is existing literature on the design of interactions for colloidal clustering. However, an experimental model with in-situ tunable SALR was never reported to the best of our knowledge. This differs from purely competing long-ranged interactions used for self-limiting assemblies, or with other assembly schemes that rely on multicomponent building blocks or directional bonding. Our interest in such SALR interactions stems from their occurrence in biological matter, especially in protein solutions, which are of both fundamental and technological relevance. At the same time, SALR is a relatively simple phenomenological model with only symmetric pair-interactions yet complex consequences in assembly. To address this point and clarify the difference with the colloidal clustering literature, we modified the following in page 17:

'This experimental model opens the door for design of clusters and other mesophases starting from simple components and using inverse methods'

With all respect to the in-situ capabilities of external magnetic fields. These forces are angle dependent due to the nature of dipolar interactions and the SALR system is highly limited to 2D because of this nature of dipolar interactions. The system authors suggest is growing clusters with highly limited control on the cluster size and arrangement and still they claim they resemble protein-protein interactions. The proteins are rather complex in nature compared to colloids and their interactions are in 3D. I believe the short range attractive and long-range attractive interactions between proteins can also be heterogeneous/directional in bonding character and the level of interactions will depend also in the salinity and pH of the solution. However, the interactions in this paper are homogenous in the plane of assembly. Therefore, I donot see much overlap between the interactions presented here and the protein-protein interactions. The depletion interactions are 3D in nature but the authors rather try to limit them in 2D in their assemblies. A reason for this is that the combination of angle dependent dipolar interactions with 3D depletion interactions are somewhat limited to 2D assemblies. I also donot agree with this statement below 'At the same time, SALR is a relatively simple phenomenological model with only symmetric pair-interactions yet complex consequences in assembly.'

The model is simple and the clusters observed are also very simple. There is no complexity in the cluster formation and there is limited control in the cluster design/formation. I donot see the complexity here. Rather small 2D aggregates with a preferential triangular shape with no description of the shape formed.

Original comment 2: The 2D crystals are forming only on the substrate or also in bulk? The depletion forces seem to be weaker than the dipolar forces therefore the assembly seems to be limited at the substrate. Please clarify this point.

Response: Our sample configuration promotes assembly in 2D purely from depletion interactions. The dipolar forces are introduced out-of-plane, i.e., perpendicular to the plane of crystal formation. Thus,

the magnetic interaction is repulsive and acts to disassemble the crystals. We changed the schematic of Figure 1 to help clarify this point.

The assembly happens in 2D and that is a limitation of the system. However, the depletion forces has a 3D nature. I was interested in a discussion about such nature of the depletion forces in combination with the angle dependent nature of the dipolar interactions. Because the dipolar magnetic interactions are angle dependent and it can be both attractive and repulsive depending on the angle between the dipoles. A SALR system the authors claim is only possible in 2D when they go out of plane configurations. They will not achieve such SALR interactions. Having a rather focused view your own capabilities and overseeing what is achieved in literature is in my view is poor science. I do not think the discussion given above addresses/ discusses what I was raising in my comment repeated above. From the discussion above one can see that my comments on the 2D vs 3D assemblies and the nature of interactions given in Comment 1 and 2 were not addressed.

I was only satisfied with the answer of the comment 3 that I have made.

On request from the editor, I have checked the comments of the Reviewer 3. These are minor comments and one (2nd) is similar to my concern asking about the interactions in the out-of-plane direction. The authors mention that the out-of-plane chaining occurs only at high magnetic field strengths and in this work they never apply such high fields. They do not elaborate nor discuss more on any out-of-plane depletion or magnetic interactions. One may like to hear more but this mostly answers the comment.

About the comment 1 of reviewer 3, authors discuss the interactions of ferrofluid particles under magnetic field and address that the dipolar interactions are not likely cause any chaining of the ferrofluid particles therefore the scenario that the Reviewer 3 addresses is not taking place and rather insignificant.

The answer to this part of the comment is sufficient. However the argument authors bring that 'it would be very interesting to manipulate the strength of the depletion attraction by changing the morphology and chaining state of the depletants in-situ.' raises a false hope. It is within the current setup not very likely. In addition, this is also not very likely to happen in general because when you chain up your depletants you are creating more free volume for the PS colloids therefore reducing the depletion. This is in a way you are reducing the concentration of the depletant NPs. I urge the authors to reconsider this point.

Response to Reviewers' comments on 'Competitive interactions via the 'dual nature' of magnetic nanoparticle dispersions'

Reviewer 1

Comment 1: The authors have addressed the review comments well. However, they should explain why their system is continuously dissipating energy, as this seems to be at the heart of the confusion surrounding terminology. Once the magnetic field is applied, where is the continuing energy source? A permanent magnet generates a magnetic field without expending energy, I think. Perhaps their magnetic field is generated by an electromagnet, which does continuously expend energy, but this is a different issue.

Response: We thank the reviewer for the comment which we have now clarified in the main text by introducing following sentence on page 4:

"The assemblies continually break, merge, and reconfigure. while dissipating external magnetic energy **input via the electromagnet**, in a unique class of *dynamic*-SALR (*d*-SALR)"

Reviewer 2

Comment 1a: With all respect to the in-situ capabilities of external magnetic fields. These forces are angle dependent due to the nature of dipolar interactions and the SALR system is highly limited to 2D because of this nature of dipolar interactions. The system authors suggest is growing clusters with highly limited control on the cluster size and arrangement and still they claim they resemble protein-protein interactions. The proteins are rather complex in nature compared to colloids and their interactions are in 3D. I believe the short range attractive and long-range attractive interactions between proteins can also be heterogeneous/directional in bonding character and the level of interactions will depend also in the salinity and pH of the solution. However, the interactions in this paper are homogenous in the plane of assembly. Therefore, I donot see much overlap between the interactions presented here and the protein-protein interactions. The depletion interactions are 3D in nature but the authors rather try to limit them in 2D in their assemblies. A reason for this is that the combination of angle dependent dipolar interactions with 3D depletion interactions are somewhat limited to 2D assemblies.

Response: We thank the reviewer for re-evaluating our work and corresponding comments. We agree that dipolar interactions are anisotropic in 3D and the protein-protein interactions are much more complex. We also agree that our SALR system is specifically applicable in the 2D plane containing the microparticles. However, our manuscript presents a model system to study SALR interactions in a controllable manner. This is only possible because we purposefully studied the assembly in 2D over 3D, which enables deconvolution of angle-dependent dipolar integrations and isotropic depletion interactions. Note that the primary purpose of this manuscript is to establish the dual nature of magnetic nanoparticle dispersions and achieve a control over the SALR interaction potential and not to direct the assembly of colloids in 3D.

We agree with all the statements made by the reviewer regarding the complexity of protein-protein interactions. Nowhere in the manuscript do we claim that our SALR represent protein-protein interactions or biological interactions. Note that the corresponding author, Bharti has extensively worked for over a decade in the area of protein adsorption and assembly and recognizes the complexity in protein-protein interactions.

Comment 1b: I also donot agree with this statement below ‘At the same time, SALR is a relatively simple phenomenological model with only symmetric pair-interactions yet complex consequences in assembly.’

The model is simple and the clusters observed are also very simple. There is no complexity in the cluster formation and there is limited control in the cluster design/formation. I donot see the complexity here. Rather small 2D aggregates with a preferential triangular shape with no description of the shape formed.

Response: In this article we only focused on the simplest case of isomerism in six particle clusters. We have not discussed higher order clusters which show non-trivial arrangements and reorganization. Even the selection of triangular over chevron and parallelogram clusters due to the long-range repulsion is a complex process. Note that no such statement on complexity of our cluster structure is made in the manuscript. The word ‘complex’ appears at four instances with reference to other works and once to the energy landscape. Regardless, we would like to again emphasize that the primary purpose of the current article is to establish the dual nature of nanoparticle dispersions and corresponding controllable SALR which is highlighted both in the title and the abstract.

Comment 2: The assembly happens in 2D and that is a limitation of the system. However, the depletion forces has a 3D nature. I was interested in a discussion about such nature of the depletion forces in combination with the angle dependent nature of the dipolar interactions. Because the dipolar magnetic interactions are angle dependent and it can be both attractive and repulsive depending on the angle between the dipoles. A SALR system the authors claim is only possible in 2D when they go out of plane configurations. They will not achieve such SALR interactions. Having a rather focused view your own capabilities and overseeing what is achieved in literature is in my view is poor science. I do not think the discussion given above addresses/ discusses what I was raising in my comment repeated above. From the discussion above one can see that my comments on the 2D vs 3D assemblies and the nature of interactions given in Comment 1 and 2 were not addressed.

I was only satisfied with the answer of the comment 3 that I have made.

Response: We agree with the reviewer that because of the angle dependent dipolar interactions, the attraction and repulsions compete in the plane i.e. are SALR type and cooperate in the direction of the field and are attractive. We disagree with the view of the reviewer opinion ‘*a rather focused view your own capabilities and overseeing what is achieved in literature is in my view is poor science*’. In our specific case, limiting the study in 2D allows us to simplify the system and understand the role of long-range repulsions in SALR fluids. Such capability was never reported before, and our system provides a new toolset to look into the structure and dynamics of the SALR fluids.

Comment 3: On request from the editor, I have checked the comments of the Reviewer 3. These are minor comments and one (2nd) is similar to my concern asking about the interactions in the out-of-plane direction. The authors mention that the out-of-plane chaining occurs only at high magnetic field strengths and in this work they never apply such high fields. They do not elaborate nor discuss more on any out-of-plane depletion or magnetic interactions. One may like to hear more but this mostly answers the comment.

About the comment 1 of reviewer 3, authors discuss the interactions of ferrofluid particles under

magnetic field and address that the dipolar interactions are not likely cause any chaining of the ferrofluid particles therefore the scenario that the Reviewer 3 addresses is not taking place and rather insignificant.

Response: We thank the author for reading our response to Reviewer 3.

Comment 4: The answer to this part of the comment is sufficient. However the argument authors bring that 'it would be very interesting to manipulate the strength of the depletion attraction by changing the morphology and chaining state of the depletants in-situ.' raises a false hope. It is within the current setup not very likely. In addition, this is also not very likely to happen in general because when you chain up your depletants you are creating more free volume for the PS colloids therefore reducing the depletion. This is in a way you are reducing the concentration of the depletant NPs. I urge the authors to reconsider this point.

Response: We are currently working on the idea of 'reducing' the effective concentration of nanoparticles in-situ using strong magnetic fields (and its gradient) and believe this is a way of altering the short-range attraction in SALR. We stand-by the statement.